# Systemic and Airway Epigenetic Disruptions Are Associated with Health Status in COPD

**DOI:** 10.3390/biomedicines11010134

**Published:** 2023-01-05

**Authors:** Ana I. Hernandez Cordero, Xuan Li, Chen Xi Yang, Julia Yang, Julia L. MacIsaac, Kristy Dever, Michael S. Kobor, Stephen Milne, Stephan F. van Eeden, Tawimas Shaipanich, Stephen Lam, Janice M. Leung, Don D. Sin

**Affiliations:** 1Centre for Heart Lung Innovation, St. Paul’s Hospital and University of British Columbia, Vancouver, BC V6Z 1Y6, Canada; 2Edwin S.H. Leong Healthy Aging Program, Department of Medical Genetics, University of British Columbia, Vancouver, BC V6T 1Z3, Canada; 3Division of Respiratory Medicine, Department of Medicine, University of British Columbia, Vancouver, BC V5Z 1M9, Canada; 4Sydney Medical School, The University of Sydney, Sydney, NSW 2050, Australia; 5British Columbia Cancer Agency, Vancouver, BC V5Z 1G1, Canada

**Keywords:** epigenetics, COPD, blood, airway, SGRQ

## Abstract

Epigenetic modifications are common in chronic obstructive pulmonary disease (COPD); however, their clinical relevance is largely unknown. We hypothesized that epigenetic disruptions are associated with symptoms and health status in COPD. We profiled the blood (*n* = 57) and airways (*n* = 62) of COPD patients for DNA methylation (*n* = 55 paired). The patients’ health status was assessed using the St. George’s Respiratory Questionnaire (SGRQ). We conducted differential methylation analyses and identified pathways characterized by epigenetic disruptions associated with SGRQ scores and its individual domains. 29,211 and 5044 differentially methylated positions (DMPs) were associated with total SGRQ scores in blood and airway samples, respectively. The activity, impact, and symptom domains were associated with 9161, 25,689 and 17,293 DMPs in blood, respectively; and 4674, 3730 and 5063 DMPs in airways, respectively. There was a substantial overlap of DMPs between airway and blood. DMPs were enriched for pathways related to common co-morbidities of COPD (e.g., ageing, cancer and neurological) in both tissues. Health status in COPD is associated with airway and systemic epigenetic changes especially in pathways related to co-morbidities of COPD. There are more blood DMPs than in the airways suggesting that blood epigenome is a promising source to discover biomarkers for clinical outcomes in COPD.

## 1. Introduction

Chronic obstructive pulmonary disease (COPD) is characterized by persistent airflow obstruction and shortness of breath [1]. This condition affects 384 million persons and is responsible for over 3 million deaths worldwide [2]. Although the pathogenesis of COPD has not been fully elucidated, it is caused by a complex interaction between environmental and genetic factors [3]. For example, cigarette smoking, which is the leading known risk factor for COPD, can alter gene expression, which may be mediated through epigenetic mechanisms. We have previously shown that the airway epithelium of COPD patients harbors a unique DNA methylation profile [4] and can alter gene expression without changing the DNA sequence. Whether these changes are local (i.e., in the small airways) or systemic (i.e., also reflected in blood) are uncertain. Moreover, their influence on patient-related outcomes such as symptoms or health status is also not known.

The St. George’s Respiratory Questionnaire (SGRQ) is a commonly used instrument, which captures the impact of disease (and its symptoms) on the quality of life of patients with COPD [5,6]. SGRQ is a 50-item questionnaire built on three domains: symptoms (frequency and severity of respiratory symptoms), activity (the effect of breathlessness on mobility and physical activity), and impact (the influence of disease on the psychosocial aspects of life). This tool is also used to assess the potential benefits of a treatment. A reduction of 4 units in the total SGRQ score is considered the minimum clinically important difference [7]. The molecular mechanisms underlying quality of life in COPD are not well understood. Here, we hypothesized that epigenetic dysregulation contributes to worsening health status in COPD patients and because COPD is a systemic disease, we also posited that blood will contain more epigenomic changes than in the airways. To investigate our hypothesis we conducted epigenome-wide differential methylation analyses to determine the association of blood and airway DNA methylation profiles with total SGRQ scores and its domains in COPD patients; we then compared the blood and airway epigenetic signatures and identified important pathways characterized by differential methylation.

## 2. Materials and Methods

### 2.1. Differential Effects of Inhaled Symbicort and Advair on Lung Microbiota (DISARM) Study Cohort

For this investigation we used the DISARM study, a 12-week randomized control trial (ClinicalTrials.gov [NCT02833480]) conducted in two hospitals in Vancouver, British Columbia, Canada (St. Paul’s Hospital and the British Columbia Cancer Agency). Institutional ethics approval was obtained from the University of British Columbia/the Providence Health Care Research Ethics Committee (H14-02277). This study has been fully described previously [8,9,10,11]. In brief, DISARM enrolled 89 stable COPD patients, and 63 of these patients reached the bronchoscopy stage of the study. The initial bronchoscopy was performed with the patient free of any inhaled corticosteroid (ICS)-based therapy for at least 4 weeks and were clinically stable for at least 8 weeks prior to the procedure. Bronchial brush samples were obtained from the 6th–8th generation airways (typically in the right or left upper lobes). Blood samples were also collected at the initial bronchoscopy visit. Spirometry was performed according to the recommendations of ATS/ERS [12]. Health-related quality of life was ascertained using the SGRQ 1-week following the bronchoscopy. Patients provided baseline demographic information including medications and comorbidities and also underwent pulmonary function tests. For the present study, we retained a total of 64 patients; of these 57 provided blood and 62 underwent bronchoscopy for DNA methylation profiling; and 55 of these patients had paired blood and brushing samples. A study diagram is shown in Appendix A.

### 2.2. DNA Methylation Profiling

For all participants, DNA extracts were obtained from peripheral blood (buffy coat fraction) and airway epithelial cell samples using the DNeasy Blood and Tissue Kit (Qiagen, Hilden, Germany). Unmethylated cytosine residues present in the DNA extracts were converted to uracil using the EZ DNA Methylation Kit (Zymo, Irvine, CA, USA). The Illumina Infinium MethylationEPIC BeadChip microarray was then used to profile 863,904 DNA methylation sites (CpG probes). All samples were profiled in one run and were randomized within the chip; this step was performed by technicians blinded to the patients’ clinical characteristics. To ensure that the blood and airway profiles were comparable we processed the data together according to previously described methods [4,13,14,15,16]. The beta values for the CpG probes were calculated as the ratio of methylation probe intensity to the overall intensity ranging from 0 (fully unmethylated) to 1 (fully methylated). CpG probes with a detection quality of *p* > 1 × 10^−10^, or contained non-CpGs, single nucleotide polymorphisms, or cross-hybridization probes were removed from the downstream analyses. Background correction, normalization, and batch correction steps were applied using the Normal-exponential out-of-band [17], Beta-Mixture Quantile Normalization [18], and ComBat [19] methods, respectively.

### 2.3. Differential Methylation Analysis

We conducted epigenome-wide association analyses based on the blood and airway epithelial cell DNA methylation profiles. For each tissue type we used the EPISTRUCTURE algorithm [20] to infer the population structure in our data. This software calculates principal components (PCs) based on CpGs that are highly correlated with a single nucleotide polymorphism to capture the genetic variability within a population. Blood cell proportions were estimated based on the deconvolution method by Houseman et al. [21] as implemented in the Horvath laboratory webtool (https://dnamage.genetics.ucla.edu/home, accessed on 10 August 2022). In addition, we conducted a PC analysis based on the entire blood DNA methylation profiles and correlated the first two PCs with blood cell proportions to assess their effect on blood DNA methylation; we included only the cell components with significant correlations with at least one PC (*p* < 0.05) in the downstream analyses, which included CD8pCD28nCD45Ran [memory and effector T cells], plasmablast [plasmablasts], CD4T [CD4 lymphocytes], NK [natural killer cells], Mono [monocytes] and Gran [granulocytes]). To identify differentially methylated positions (DMPs) associated with SGRQ scores, we used a robust linear model (rlm) function implemented in the MASS R package [22]. For these analyses, the beta values were logit transformed to M-values (normal distribution). Additional covariates (i.e., baseline age, sex, body mass index and smoking status) were included according to the algorithm as outlined by Lee et al. [4,14,23]. Considering the number of variables in our model, the limited sample size of our cohort and the fact that baseline FEV_1_% predicted was significantly correlated with SGRQ total score (R = −0.47, *p* = 0.001), we did not adjust our analyses for lung function. Our model for the blood cell DNA methylation profiles was defined as:Methylation (M values)~ SGRQ total score+Age+Sex+BMI+Smoking status+CD8pCD28nCD45RAn + PlasmaBlast + CD4T + NK + Mono + Gran+EPISTRUCTURE (PC1 to PC5)

The differential analysis for airway epithelial cells profiles was defined as: Methylation (M values) ~ SGRQ total score+Age+Sex+BMI+Smoking status+ EPISTRUCTURE (PC1 to PC5)

These analyses were conducted to assess the association between DNA methylation in the two tissues and SGRQ scores and its domains (Appendix A). Significant DMPs were defined based on a false discovery rate (FDR) cut-off of <0.10. We later used the R package DMRcate [24] to identify differentially methylated regions (DMR), which were defined with at least three consecutive CpGs.

### 2.4. Pathway Enrichment Analysis

We used the software package WebGestaltR to identify Kyoto Encyclopedia of Genes and Genomes (KEGG) pathways enriched by the genes characterized by differential methylation in blood and airway epithelial cells. Significant enrichment was defined at *FDR* < 0.05.

## 3. Results

### 3.1. Study Cohort Overview

An overview of the study cohort is presented in Table 1. The participants included a total of 64 adults, of these 57 were profiled for blood and 62 had airway samples; the majority of whom were males.

### 3.2. Blood and Airway Epigenetic Disruptions Are Associated with SGRQ Scores

Differential blood DNA methylation analysis based on DNA methylation profiles yielded 29,211 DMPs associated with total SGRQ scores (Figure 1A, Appendix A). These DMPs were within the vicinity of 13,485 unique genes. Table 2 shows that the top five DMPs identified in blood were located within the *ARFGAP1, RP11-711C17.2, PPARG* and *MGAT4C* genes (Figure 1A). In blood, we identified 3250 DMRs associated with total SGRQ scores (Appendix A); Table 3 shows the top five DMRs. Differentially methylated genes for total SGRQ score were enriched in 119 pathways (Figure 2A, Appendix A), including cancer pathways (e.g., small and non-small cell lung cancer), age-related pathways (e.g: longevity regulating pathway and mTOR signaling pathway), and neurological pathways (e.g., cholinergic synapse and dopaminergic synapse). The SGRQ activity, impact, and symptom domains were associated with 9161, 25,689 and 17,293 DMPs in blood, respectively. Activity score DMPs corresponded to 5508 unique genes that enriched 19 pathways (e.g., pathways in cancer, longevity regulating pathway and oxytocin signaling pathway); impact score DMPs corresponded to 11,901 genes that enriched 115 pathways (e.g., pathways in cancer, MAPK signaling pathway and PI3K-Akt signaling pathway); and symptom score DMPs were located within 8332 genes that enriched 75 pathways (e.g., platelet activation, Inflammatory mediator regulation of TRP channels and cortisol synthesis and secretion). Furthermore, 2087 genes were exclusive to activity DMPs, 6181 genes to impact DMPs and 3581 genes to symptom DMPs (Appendix A). The top genes characterized by differential methylation in blood for each SGRQ domain are shown in Table 2, and included *CCDC30, DOCK2* and *F2* for the activity score DMPs, *ERC2, AC004041.2, RAD50*, and *AP4S1* for the impact score DMPs, and *BACH2* and *WNK2* for the symptom score. We also found 1048, 2925, and 1924 DMRs for activity, impact, and symptom score, respectively; Table 3 shows the top five DMRs.

Airway differential DNA methylation was associated with SGRQ scores, albeit less than in the blood. For instance, 5044 DMPs were associated with total SGRQ score (Figure 2B), which corresponded to 2950 unique genes that enriched 38 pathways (e.g., small and non-small lung cancer, mTOR signalling pathway and insulin resistance). In addition, we identified 643 DMRs for total SGRQ score in the airway; top five DMRs are shown in Table 3. For the SGRQ activity score, we identified 4674 DMPs located within 2847 genes, which enriched 36 pathways (e.g., T and B cell receptor signaling pathways, and longevity regulating pathway); DMPs were grouped into 590 DMRs. For the impact score, we identified 3730 DMPs within 2198 genes, which enriched 8 pathways (e.g., Insulin signaling pathway, and mTOR signaling pathway); in addition we also identified 473 DMRs for the impact score. The symptom score was associated 5063 DMPs, these were located within 2850 genes that significantly enriched 24 pathways (e.g., platelet activation and cortisol synthesis and secretion); furthermore 625 DMRs were identified for the symptom score. In addition, we found that 2039 genes were unique to activity score DMPs, 1257 genes to impact score DMPs, and 1919 genes to symptom score DMPs (Appendix A). 

### 3.3. A Systemic Epigenetic Signature of Health Status in COPD 

We compared differentially methylated genes (DMGs) in blood to those in the airway epithelium. For DMGs associated with total SGRQ score, there were 1590 overlapping genes, which represented 54% of the airway epithelial DMGs (Figure 3A). These genes enriched 25 pathways (Figure 3E), most of these were captured by the blood differential methylation analyses (24 pathways). These pathways included many aging (e.g., PI3K-Akt signaling pathway, longevity regulating pathway, and Ras signaling pathway) and cancer (e.g., non-small and small cell lung cancer) pathways. For DMGs associated with SGRQ domain scores, there were 779, 1154 and 1156 overlapping genes for activity, impacts and symptoms scores, respectively, representing 27%, 53%, and 41% of the airway epithelial DMGs, respectively (Figure 3B–D). 

We also compared pathways enriched for DMGs between blood and airway epithelium. For total SGRQ score, 36 out of the 38 pathways identified in the airway epithelium (Figure 2B) overlapped with those in blood, including small and non-small cell lung cancer and mTOR signaling pathways. For SGRQ activity score, 9 out of the 36 pathways identified in the airway overlapped with those in blood (e.g., pathways in cancer and longevity regulating pathway); for SGRQ impact score, 7 out of the 8 pathways identified in the airway epithelium overlapped with those identified in blood (e.g., mTOR signaling pathway and insulin signaling pathway); for SGRQ symptom score, we identified 20 out of 24 pathways in the airway epithelium that overlapped with blood pathways, including platelet activation, pathways in cancer and Wnt signaling pathway.

## 4. Discussion

To our knowledge this the first report that directly evaluated blood and airway epigenetic signatures in relation to health status of patients with COPD. We made several novel observations. First, there are distinct epigenetic signatures that relate to health status and symptoms of patients with COPD. These signatures (both in blood and in the airway) are enriched in pathways related to accelerated ageing and lung cancer, which are important consequences and comorbidities of COPD [25]. Second, although blood carries most of the epigenetic changes observed in the airways, it also harbors distinct non-airway related epigenetic changes, which may reflect the systemic nature of COPD [26]. 

Epigenome-wide disruptions have been associated with COPD [4]; however their clinical impact has not been well characterized. Our findings suggest that differential methylation in blood and airway epithelial cells is associated with patient symptoms and health status in COPD and that these changes can be detected in blood as well as airway samples. Our analyses also highlight several interesting differentially methylated genes, which may have plausible effects in COPD. One of the most significant genes in the blood differential analyses was *PPARG*, where increased methylation within *PPARG* in the airways was associated with increased score in the SGRQ symptoms domain. This gene has anti-inflammatory functions in various cells in the lung including: airway epithelial cells, endothelial cells, airway smooth muscle cells, alveolar macrophages and eosinophils [27,28,29,30,31,32,33,34,35]. Mice experiments have shown that activation of *PPARG* downregulates the expression of inflammatory chemokines and mitigates cigarette-smoking induced emphysema [36]. COPD is related to both airway and systemic inflammation [37,38]. We found that some of the genes in the inflammatory pathways (e.g., *STAT3*, *PIAS3, IL8* [blood—all DMPs are hypermethylated], *IL6* [blood—all DMPs are hypomethylated], and *IL6R* [airway and blood—specific DMPs are hypo- or hypermethylated], *IL10* [blood—all DMPs are hypomethylated]) were differentially methylated and related to health status of our patients. In support of these findings, in vitro model has linked hypomethylation of promoters in the *NF-κB* and *STAT3* genes with the induction of inflammation by lipopolysaccharide and cigarette smoke extract [39]. 

Multiple pathways were enriched by the differentially methylated genes in both blood and airways. Overlapping pathways included age-related processes such as mTOR signaling pathway, which regulates cell proliferation, apoptosis, and autophagy in the cellular senescence process [40] and the PI3K-Akt signaling pathway. Akt activation, which occurs during ageing, may be responsible for neuronal dysfunction of ageing [41]. Akt is also involved in the regulation of mTOR [42]. In addition, cancer pathways were identified in all our enrichment analyses (e.g., lung cancer). It is well established that COPD is a major risk factor for lung cancer, increasing its risk by 2 to 4 fold [43]. Our findings suggest that methylation changes in the genome may be responsible for some of this excess risk, however more research is needed to define the effect of hypo- and hyper- methylated genes on cancer risk. Our analyses also highlighted neurological pathways, for example, dopaminergic synapse, which plays a central role in the control of behavioral processes such as addiction and stress [44]. Furthermore, dopamine has been associated with improvement of diaphragmatic function in COPD patients [45], thus its regulation may also affect respiratory symptoms. We also found that there was epigenomic changes in pathways for cholinergic signaling, which dampens inflammation by downregulating pro-inflammatory cytokines (i.e., TNF-a, IL-1B and IL-6) [46,47]. Thus, DNA methylation in COPD may not only affect physical manifestations of COPD, but also contributes to the individuals’ ability to cope with psychosocial stress of COPD. 

Our analyses were limited by several factors. First, our study cohort was small, and lacked significant sex/gender and ethnic diversity; thus our findings may not be fully generalizable to patients in the community. In addition the small sample size limited our investigation of methylation patterns in smokers and ex-smokers separately. Second, longitudinal effects of epigenetic disruptions could not be captured due to the cross-sectional methodology of DISARM. Third, we were not able to establish whether poor health status in COPD caused the epigenetic disruptions or vice versa. Fourth, due to the invasive nature of the bronchoscopy procedure, a replication cohort was not available, and thus future efforts should aim to replicate our findings. Fifth, bronchial brush samples, while mostly epithelial cells, might have also included inflammatory cells, which may could have impacted our results. However, previous research has shown that epithelial cells are the main component of bronchial brush samples [4]; likewise, although our blood analysis were adjusted for cell composition [48] blood samples have a complex mixture of immune cells, which varies across patients [49] and therefore we were not able to identify epigenetic differences for each specific cell populations. In summary, there are epigenetic disruptions in blood and airways associated with SGRQ scores, which may contribute to age-, cancer- and neurological-related processes in COPD. Our findings support the notion that the processes disrupted in the lung of COPD patients could have systemic effects that may impact their quality of life and symptoms, and that blood DNA methylation features are sensitive indicators of similar changes in the airways. Together these data suggest that blood DNA methylation patterns can be cultivated as potential biomarkers of health status and outcomes of patients with COPD. 

## Figures and Tables

**Figure 1 biomedicines-11-00134-f001:**
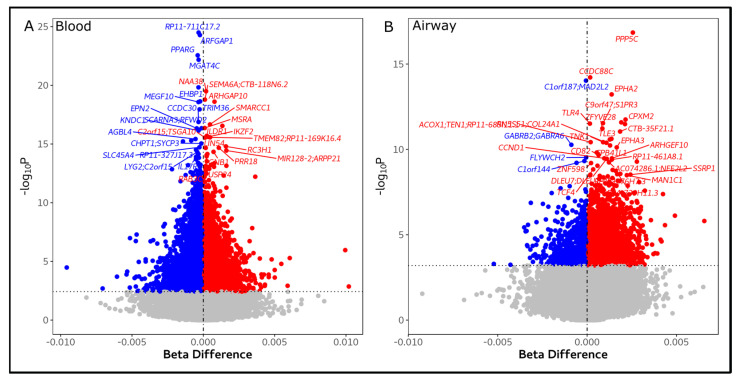
Blood (**A**) and airway (**B**) differentially methylated sites association with Total SGRQ score. x-axis represents the robust linear model (rlm) estimated effects on the methylation Beta-values, y-axis represents the rlm −log10P on M-values. For each unit of SGRQ increased, DNA methylation decreases (hypomethylation = blue) or increases (hypermethylation = red).

**Figure 2 biomedicines-11-00134-f002:**
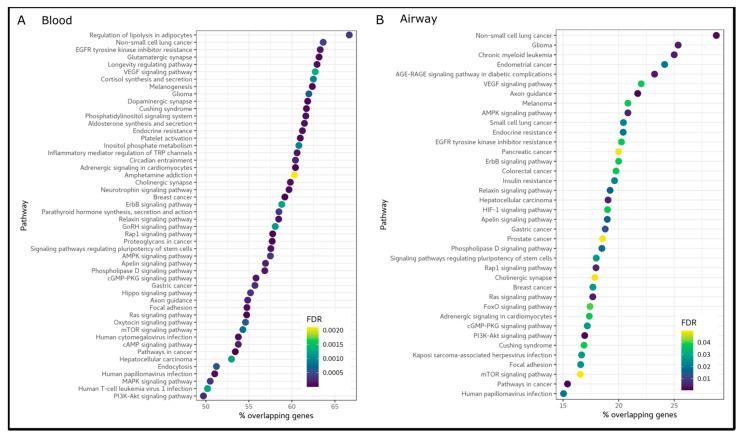
KEGG pathways enriched for differentially methylated genes for total SGRQ score. Horizontal and vertical axis represent the percentage of genes within each pathway that are characterized by differential methylation and description of the pathways, respectively. (**A**) Blood. (**B**) Airway.

**Figure 3 biomedicines-11-00134-f003:**
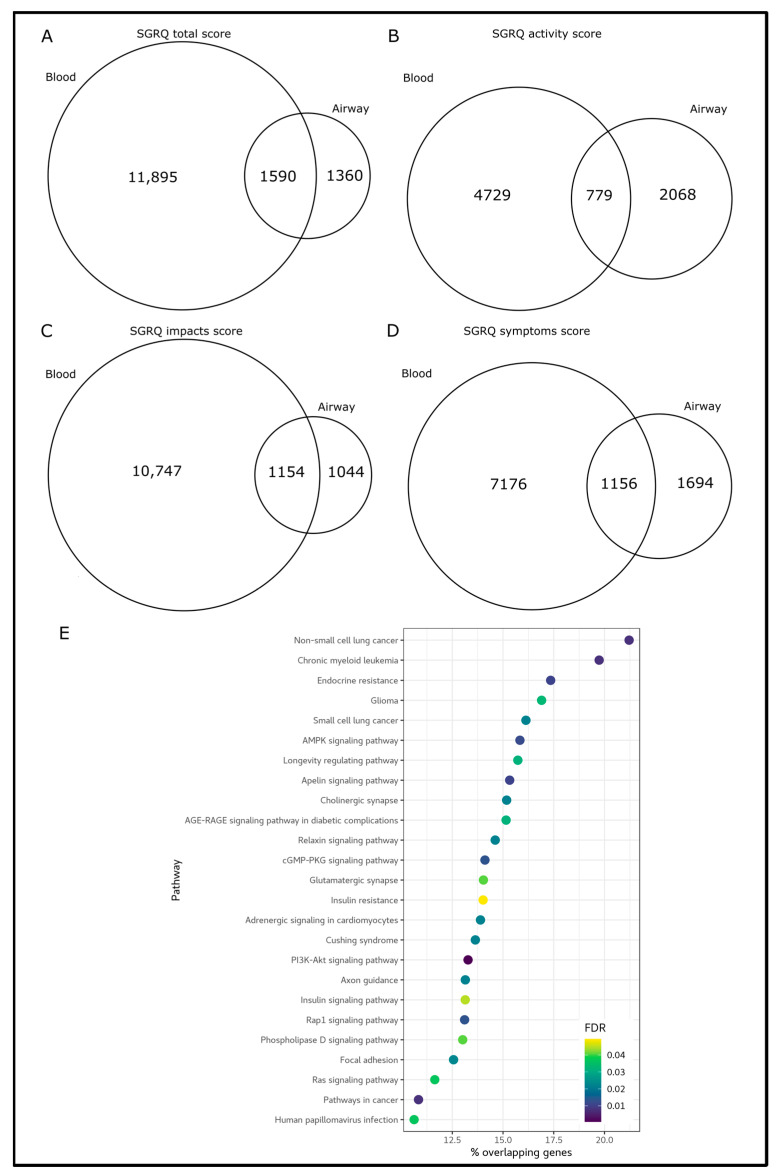
Differentially methylated genes overlapped between tissues. Venn diagrams show the overlap of differentially methylated genes identified in blood versus those in the airway for the total SGRQ score (**A**), and its domains: activity (**B**), impact (**C**) and symptom (**D**) scores. (**E**) shows the pathways enriched by the differentially methylated genes identified in blood and airway tissue.

**Table 1 biomedicines-11-00134-t001:** Demographic characteristics at baseline.

DISARM Study Cohort	
*n*	64
Age, years	64 ± 8
Female, %	17
BMI, kg/m^2^	24.58 (21.09–29.35)
Smoking status	
*Current, %*	45
*Former, %*	55
Pack per years	48.00 (33.00–59.50)
SGRQ total score	44.06 (33.08–54.90)
SGRQ activity score	65.89 (48.52–74.08)
SGRQ impacts score	28.45 (18.10–39.68)
SGRQ symptoms score	56.20 (39.30–70.63)
FEV_1_% of predicted	55.00 (43.65–67.25)
FVC% of predicted	83.35 (71.90–92.25)
FEV_1_/FVC, percent	52.64 (42.93–60.09)

BMI: body mass index. FEV1: forced expiratory volume in 1 s. FVC: Forced vital capacity. Age is shown as mean and standard deviation. Count data are shown in percentages, and Lung function variables are reported as median and interquartile range as variables are not normally distributed.

**Table 2 biomedicines-11-00134-t002:** Top five blood and airway DMPs for SGRQ scores.

SGRQ Score	Tissue	Probe	Beta Difference	*p*	*FDR*	Chr	Relation to Island	Position in Relation to Gene	Gene Symbol
Total	Blood	cg00542760	−0.0002	5.22 × 10^−25^	2.06 × 10^−19^	20	Island	5’UTR; 1stExon; TSS1500; 3’UTR	*ARFGAP1*
		cg02344187	−0.0003	3.21 × 10^−25^	2.06 × 10^−19^	12	Open Sea	5’UTR	*RP11-711C17.2*
		cg25911248	−0.0004	2.78 × 10^−23^	7.32 × 10^−18^	3	Open Sea	3’UTR	*PPARG*
		cg00151915	−0.0003	6.59 × 10^−23^	1.30 × 10^−17^	12	Open Sea	5’UTR	*MGAT4C*
		cg02213440	−0.0003	1.49 × 10^−20^	2.35 × 10^−15^	7	Open Sea		
	Airway	cg16929656	0.0026	1.44 × 10^−17^	1.13 × 10^−11^	19	Open Sea	3’UTR	*PPP5C*
		cg05245430	0.0002	6.08 × 10^−15^	2.40 × 10^−09^	14	Island	3’UTR; 1stExon	*CCDC88C*
		cg21153875	−0.0001	9.40 × 10^−15^	2.47 × 10^−09^	1	Island	TSS200	*C1orf187; MAD2L2*
		cg15346134	0.0014	6.01 × 10^−14^	1.18 × 10^−08^	1	Open Sea	TSS200	*EPHA2*
		cg15550234	0.0021	1.78 × 10^−12^	2.81 × 10^−07^	10	Open Sea	3’UTR; 5’UTR	*CPXM2*
Activity	Blood	cg09711814	−0.0001	4.35 × 10^−22^	3.43 × 10^−16^	7	Open Sea		
		cg24639069	−0.0002	1.76 × 10^−18^	6.94 × 10^−13^	1	Open Sea	TSS1500; 5’UTR	*CCDC30*
		cg10677105	−0.0005	6.97 × 10^−15^	1.83 × 10^−09^	5	Open Sea	5’UTR; 3’UTR	*DOCK2*
		cg11893552	0.0015	1.86 × 10^−14^	3.66 × 10^−09^	6	Open Sea		
		cg00371195	−0.0001	3.40 × 10^−14^	4.47 × 10^−09^	11	Open Sea	TSS1500	*F2*
	Airway	cg00278597	0.0002	4.35 × 10^−12^	1.71 × 10^−06^	8	Open Sea	TSS1500	*RP11-1057B8.2*
		cg09397653	0.0009	2.60 × 10^−12^	1.71 × 10^−06^	15	Open Sea	TSS1500	*ITGA11*
		cg27547307	0.0005	1.11 × 10^−11^	2.91 × 10^−06^	17	Open Sea	TSS1500; 5’UTR	*CYTH1*
		cg00413620	0.0011	1.74 × 10^−11^	3.43 × 10^−06^	1	Open Sea		
		cg04926227	0.0011	4.02 × 10^−11^	6.33 × 10^−06^	8	Open Sea	TSS1500; 3’UTR; 5’UTR	*RP11-463D19.1; STAU2*
Impact	Blood	cg23444468	0.0004	6.72 × 10^−30^	5.29 × 10^−24^	3	Island	TSS1500; 5’UTR	*ERC2*
		cg13886298	−0.0002	1.45 × 10^−24^	5.70 × 10^−19^	5	Open Sea	TSS1500; 3’UTR	*AC004041.2; RAD50*
		cg15751204	0.0008	5.45 × 10^−22^	1.43 × 10^−16^	3	Open Sea		
		cg00851837	−0.0003	4.31 × 10^−20^	8.48 × 10^−15^	14	Open Sea	TSS200	*AP4S1*
		cg15534855	0.0005	1.24 × 10^−19^	1.95 × 10^−14^	18	Island		
	Airway	cg08738303	0.0005	3.69 × 10^−13^	2.91 × 10^−07^	18	Open Sea		
		cg01585096	0.0012	3.64 × 10^−12^	7.16 × 10^−07^	19	Open Sea	TSS200; 3’UTR	*KRTDAP*
		cg03053018	0.0030	2.11 × 10^−12^	7.16 × 10^−07^	7	Open Sea		
		cg20447038	0.0018	3.22 × 10^−12^	7.16 × 10^−07^	6	Open Sea		
		cg02065293	0.0011	1.08 × 10^−11^	1.46 × 10^−06^	2	Open Sea		
Symptom	Blood	cg06894541	0.0003	3.10 × 10^−19^	2.44 × 10^−13^	2	Open Sea		
		cg25670076	0.0021	4.09 × 10^−18^	1.61 × 10^−12^	6	Open Sea	5’UTR; 3’UTR	*BACH2*
		cg11743078	−0.0004	1.87 × 10^−15^	4.92 × 10^−10^	1	Open Sea		
		cg02415617	−0.0004	2.08 × 10^−14^	3.25 × 10^−09^	9	South Shore	1stExon; 3’UTR; 5’UTR	*WNK2*
		cg04028140	0.0007	2.48 × 10^−14^	3.25 × 10^−09^	11	Open Sea		
	Airway	cg07380540	−0.0010	6.04 × 10^−26^	4.76 × 10^−20^	1	South Shelf		
		cg10789584	0.0005	2.76 × 10^−17^	1.09 × 10^−11^	11	Open Sea	5’UTR	*CD82*
		cg18910215	−0.0007	5.49 × 10^−14^	1.44 × 10^−08^	9	Open Sea	5’UTR	*MAPKAP1*
		cg20708037	0.0018	9.37 × 10^−14^	1.85 × 10^−08^	1	Open Sea		
		cg21088488	0.0008	4.81 × 10^−12^	7.58 × 10^−07^	7	South Shore	3’UTR; TSS1500	*DBNL; PGAM2*

Top CpGs criteria: smallest to largest FDR. Beta difference was estimated from each methylation site beta value and *p*-value was estimated from each methylation site M-value. Negative beta: for each unit of SGRQ increased, DNA methylation decreases. Positive beta: for each unit of SGRQ increased DNA methylation increases.

**Table 3 biomedicines-11-00134-t003:** Top five blood and airway DMRs for SGRQ scores.

SGRQ Score	Tissue	Chr	Start	End	# CpGs	Min *FDR*	Gene Symbols
Total	Blood	3	47,823,638	47,825,578	7	4.71 × 10^−27^	*SMARCC1*
		12	124,246,976	124,248,926	5	8.50 × 10^−24^	*DNAH10*
		20	61,917,085	61,918,367	5	9.22 × 10^−23^	*ARFGAP1, MIR4326*
		4	148,653,624	148,654,701	5	7.03 × 10^−20^	*ARHGAP10*
		5	126,779,737	126,780,974	4	1.47 × 10^−19^	*MEGF10*
	Airway	18	56,296,094	56,296,607	10	2.90 × 10^−22^	*ALPK2, RPL9P31*
		9	91,604,473	91,605,695	7	5.88 × 10^−18^	*C9orf47, S1PR3*
		19	46,894,811	46,895,714	3	1.41 × 10^−16^	*AC007193.8*
		12	11,698,534	11,699,363	5	3.03 × 10^−14^	*RP11-434C1.1, RNU7-60P*
		1	120,173,989	120,175,029	7	3.03 × 10^−14^	
Activity	Blood	11	2,019,436	2,021,103	32	9.13 × 10^−19^	*H19*
		6	168,045,268	168,046,457	6	4.18 × 10^−18^	
		3	30,936,070	30,936,955	11	4.98 × 10^−14^	*GADL1*
		17	699,291	700,672	4	5.21 × 10^−13^	
		7	94,285,270	94,287,242	60	1.11 × 10^−12^	*SGCE, PEG10*
	Airway	18	56,296,094	56,296,607	10	5.98 × 10^−23^	*ALPK2, RPL9P31*
		11	86,085,026	86,086,489	12	1.24 × 10^−13^	*CCDC81*
		19	29,217,858	29,218,774	7	1.31 × 10^−11^	*AC005307.3*
		18	3,411,487	3,412,713	11	1.37 × 10^−11^	*TGIF1*
		7	157,866,683	157,868,361	13	2.33 × 10^−11^	
Impact	Blood	6	42,927,199	42,928,920	31	6.89 × 10^−33^	*GNMT*
		3	56,501,352	56,502,814	12	1.26 × 10^−30^	*ERC2*
		18	74,960,629	74,963,364	35	6.35 × 10^−27^	*GALR1*
		10	134,598,316	134,601,851	37	4.30 × 10^−24^	*NKX6-2, RP11-288G11.3*
		3	47,823,674	47,825,578	6	1.03 × 10^−23^	*SMARCC1*
	Airway	19	35,981,224	35,982,442	10	1.20 × 10^−19^	*KRTDAP*
		17	75,470,567	75,472,168	12	2.22 × 10^−15^	*SEPT9, RP11-75C10.9*
		21	43,315,518	43,316,705	6	9.74 × 10^−14^	
		12	11,698,367	11,699,363	6	2.63 × 10^−13^	*RP11-434C1.1, RNU7-60P*
		3	113,160,071	113,161,177	14	1.70 × 10^−12^	*WDR52*
Symptom	Blood	6	32,807,895	32,811,521	30	3.27 × 10^−24^	*PSMB8, TAP2, PSMB9, TAPSAR1*
		5	78,364,769	78,366,302	14	2.24 × 10^−22^	*DMGDH, BHMT2*
		6	30,850,207	30,852,354	24	2.24 × 10^−22^	*DDR1*
		2	110,969,641	110,970,909	8	4.86 × 10^−22^	*LINC00116*
		2	98,329,337	98,330,493	10	1.96 × 10^−21^	*ZAP70*
	Airway	6	33,244,976	33,246,895	44	1.04 × 10^−22^	*B3GALT4, WDR46, RPS18*
		12	63,025,490	63,026,424	7	2.02 × 10^−21^	
		9	139,425,582	139,427,171	5	5.63 × 10^−20^	
		17	46,655,164	46,656,572	20	6.39 × 10^−17^	*HOXB4, MIR10A, HOXB3*
		2	161,992,157	161,993,364	6	1.19 × 10^−16^	*TANK*

Top differentially methylated region criteria: smallest to largest minimum FDR.

## Data Availability

Data available upon request to authors (D.D.S., A.H.C.)

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
