# Peer review of "Systemic and Airway Epigenetic Disruptions Are Associated with Health Status in COPD"

_biomedicines, 2023, doi:10.3390/biomedicines11010134_

Round 1
Reviewer 1 Report
Hernandez Cordero and coll. conducted an interesting epigenome-wide study on blood and airway DNA samples obtained from patients with COPD enrolled in the DISARM trial. They put in relation the results of DNA methylation analysis with patients’ overall health status as assessed by the St. George Respiratory Questionnaire (which reflected spirometry data). They found in both types of tissues epigenetic signatures enriched in pathways correlated with inflammation, accelerated ageing, lung cancer and neurological disruptions. The patients in the study were free from any corticosteroid therapy in the preceding 4 weeks, thus ensuring the exclusion of a potential bias on the measurements done. The proposed use of blood DNA methylation as potential biomarker of health status in COPD clearly requires further validation in larger cohorts.
While the study is overall clearly designed and conducted, I found it difficult to understand how the mathematical model used for differential methylation analysis was constructed. If possible, please Authors try to make this section easier to read.
Major points:
- Did the authors explore the integration of EWAS with GWAS? In other terms, did the observed DMPs colocalise with any SNPs, and specifically COPD-associated SNPs? This may provide a hint about the relative influence of genetic or environmental factors in determining COPD health status and may highlight differences in the two samples considered.
- Did the authors test any possible correlations between DNA methylation and mRNA gene expression? This could be important to infer potential functional effects of differentially methylated positions in either blood or airway samples and to correlate them with patients’ health status.
Minor points:
- Please add SGRQ scores (mean, range) to Table I.
Reviewer 2 Report
The authors of paper conducted epigenome-wide differential methylation analyses to determine the association of blood and airway DNA methylation profiles with total SGRQ scores and its domains in COPD patients.
The paper s interest, but it’s difficult for readers to understand.
Comments.
Is it possible to influence on blood DNA methylation patterns? For example for treatment of COPD.
The authors must check references.
Reviewer 3 Report
With interest, I read the paper biomedicines-2080637. It is written by experienced researchers.
I have only a few comments:
1. The Authors report that they used buffy coats, which means that mixed cell populations were used. Even though the Authors implemented appropriate bioinformatic correction tools, this should be mentioned and briefly discussed as a study limitation, compared to the studies using sorted cells (PMID: 28322581, 24495553).
2. Likewise, the Authors report using airway epithelial cell samples obtained with a bronchoscopy brushing method. What was the purity of those cells? How was it verified? How were potential contaminants eliminated? Please, address.
3. Please, make sure that the names of the genes are always written in italics, including tables and figures.
